# Filming ultrafast roaming-mediated isomerization of bismuth triiodide in solution

Eun Hyuk Choi [1,2], Jong Goo Kim[1,2], Jungmin Kim[1,2], Hosung Ki[1,2], Yunbeom Lee[1,2], Seonggon Lee[1,2], Kihwan Yoon[3], Joonghan Kim[3], Jeongho Kim[4] & Hyotcherl Ihee [1,2 ✉]

Roaming reaction, defined as a reaction yielding products via reorientational motion in the long-range region (3 – 8 Å) of the potential, is a relatively recently proposed reaction pathway and is now regarded as a universal mechanism that can explain the unimolecular dissociation and isomerization of various molecules. The structural movements of the partially dissociated fragments originating from the frustrated bond fission at the onset of roaming, however, have been explored mostly via theoretical simulations and rarely observed experimentally. Here, we report an investigation of the structural dynamics during a roaming-mediated isomerization reaction of bismuth triiodide ($BiI_3$) in acetonitrile solution using femtosecond time-resolved x-ray liquidography. Structural analysis of the data visualizes the atomic movements during the roaming-mediated isomerization process including the opening of the $Bi-I_b-I_c$ angle and the closing of $I_a-Bi-I_b-I_c$ dihedral angle, each by ~40°, as well as the shortening of the $I_b{\cdots}I_c$ distance, following the frustrated bond fission.

[1] Department of Chemistry and KI for the BioCentury, Korea Advanced Institute of Science and Technology (KAIST), Daejeon, Republic of Korea. [2] Center for Nanomaterials and Chemical Reactions, Institute for Basic Science (IBS), Daejeon, Republic of Korea. [3] Department of Chemistry, The Catholic University of Korea, Bucheon, Republic of Korea. [4] Department of Chemistry, Inha University, Incheon, Republic of Korea. ✉email: hyotcherl.ihee@kaist.ac.kr

Since its discovery from the unimolecular dissociation of formaldehyde in the gas phase, the roaming reaction has attracted much attention for its unusual reaction products and their energy distributions, which are difficult to understand with the traditional transition state theory, presuming that the products are generated following the minimum energy pathway[1–10]. Subsequent experimental and theoretical studies have found that roaming reactions are not restricted to specific molecules and now have been generally accepted as a universal mechanism for many unimolecular decompositions and isomerization reactions in the gas phase[11–15]. The dissociation of formaldehyde yielding molecular $H_2$ and CO is known as the textbook example of roaming reactions[1,16], and the isomerization of nitrobenzene[17] and nitromethane[12] in their decomposition reactions is also known to occur via roaming, which has been termed roaming-mediated isomerization.

All these findings on roaming reactions were obtained from the gas-phase reactions, but a recent study on the photoinduced isomerization of geminal tri-bromides in solution has suggested that the isomer formation of the geminal tri-bromides is compatible with roaming-mediated isomerization, indicating roaming reactions are working in the liquid solution phase as well[15]. In that study, various geminal tri-bromides, $XBr_3$ (X = CH, B, or P), were investigated, and it was found that the formation of the isomer, (Br-X-Br-Br), occurs on the sub-100 fs time scale. Based on the similar photodynamics of those tri-bromides and their shared topological features of potential energy surfaces including the $S_1/S_0$ conical intersection and a flat region on the $S_1$ surface, it was suggested that the roaming-mediated ultrafast isomerization might be a general reaction pathway regardless of the nature of the coordination center. Thus, these findings indicate that through the roaming on the flat region of the potential, an efficient formation of molecular products such as an isomer can occur upon a photofragmentation reaction in the solution phase as in the gas phase.

Most of the previous studies on roaming reactions have been conducted using velocity map imaging techniques or optical spectroscopies. Even though those experiments have provided a profound understanding of the molecular energetics of a reaction, they do not give direct information on the change of molecular structure. The nuclear trajectories of the reacting molecules during a roaming reaction have been investigated only via sophisticated theoretical simulations. As one of the direct structural probes[18–26], time-resolved X-ray liquidography (TRXL) can be a suitable tool to tackle this issue[27–42]. The temporal resolution of TRXL now reaches the femtosecond time domain with the recent advent of X-ray free-electron lasers (XFELs) providing ultrashort (typically several tens of femtoseconds) hard X-ray pulses with high photon fluxes. In particular, femtosecond-TRXL (fs-TRXL) has allowed us to track the nuclear positions frame by frame, which can be considered wavepacket trajectories[33,38–40,43,44]. For example, the wavepacket trajectories for chemical bond formation and intramolecular vibrations of a gold trimer complex in solution have been observed using fs-TRXL[41,42].

Bismuth triiodide ($BiI_3$) is a geminal tri-halide with a trigonal pyramidal structure with Bi incorporated as the center atom and has been known to undergo dissociation upon UV-visible irradiation[45]. A TRXL study on the photodynamics of $BiI_3$ in acetonitrile solution has shown that the photoreaction of $BiI_3$ upon 400 nm excitation proceeds via two parallel reaction pathways, (i) dissociation to radical fragments, $BiI_2\bullet$ and $I\bullet$, and (ii) isomerization to $iso\text{-}BiI_2\text{-}I$[37] (Fig. 1). The isomer of $BiI_3$ belongs to the same class as the isomers of the geminal tri-bromides as well as previously reported isomers of halocarbons such as $CHI_3$[46,47], $CH_2I_2$[48], $CH_2BrI$[49], and $CF_2I_2$[50]. This type of isomer is generally known to be formed by the radical recombination

following the collision with the solvent cage, which is cage-induced isomerization, in several hundreds of femtoseconds to tens of picoseconds. However, as in the case of geminal tri-bromides, when the isomer of $BiI_3$ is formed via direct intramolecular rearrangement in the time range much faster than the time for solute-solvent interaction, the formation of the isomer of $BiI_3$ can also be regarded as the roaming-mediated isomerization. Since the atoms constituting $BiI_3$ are all heavy, every atom pair significantly contributes to the time-resolved difference scattering signals (see Supplementary Fig. 1), and therefore, $BiI_3$ serves as an excellent model system for tracking the molecular structure undergoing isomerization.

In this article, we report the results of fs-TRXL experiment on $BiI_3$ dissolved in acetonitrile. In particular, we determined the kinetics of both isomerization and dissociation reaction pathways as well as the structures of the transient species formed during the reaction. We observe that a precursor isomer whose structure is slightly different from the previously found isomer (late isomer) is formed at the early stage of the reaction, and it acts as the doorway to the late isomer generated at late time delays and the ground-state $BiI_3$. The isomerization from the ground-state $BiI_3$ to the early isomer is completed in about 100 fs, which coincides with the expected timescale for a roaming-mediated isomerization. We also visualize the structural movements involved in the isomerization as well as the dissociation of $BiI_3$.

## Results and discussion

**Experimental isotropic fs-TRXL signal $\Delta S_{iso}(q, t)$.** The time-resolved difference scattering images of the $BiI_3$ solution, which are anisotropic (see Supplementary Fig. 2), were decomposed into isotropic and anisotropic contributions of the scattering signals via a well-established method[51,52]. Comparing the anisotropic contributions of difference scattering signals of $BiI_3$ solution to those of a dye solution (see Supplementary Fig. 3), which contain transient signals only from the solvent response, shows that they are identical. This result means a negligible contribution from $BiI_3$ solute molecules to the anisotropic component of the scattering signals. Hence, we focus on the isotropic part, $\Delta S_{iso}(q, t)$, of the difference scattering signals of the $BiI_3$ solution. The details of the methods for the experiment and data processing are described in the "Methods" section as well as in Supplementary Methods and their procedures are summarized in Supplementary Fig. 4.

The isotropic contribution, $\Delta S_{iso}(q, t)$, shown in Fig. 2a evolves over a broad time range, with its shape and intensity changing. In particular, the negative and positive peaks in the entire $q$ range shift toward lower $q$ in less than ~500 fs, followed by a significant reduction of the overall amplitude of the signal with a slight change of the peak positions within ~10 ps. These qualitative features revealed by visual inspection imply that the signal may contain the information on the ultrafast motion of excited solute molecules, and the majority of the excited molecules relax back to the ground state within 10 ps.

**Kinetics and equilibrium geometries of reacting species.** The $\Delta S_{iso}(q, t)$ of $BiI_3$ solution has rich features, including the signals from both exponential kinetics and coherent dynamics (Fig. 2a). The singular value decomposition (SVD) of $\Delta S_{iso}(q, t)$ shows four significant singular vectors (see Supplementary Fig. 5). To simplify the analysis, we first extracted the signal presumably corresponding to the exponential kinetics by filtering out the third and fourth singular vectors and we termed the extracted signal shown in Fig. 2d as $\Delta S_{iso}'(q, t)$, as detailed in "Methods" and Supplementary Methods. The first two right singular vectors (RSVs), which contain the overall exponential kinetics, were fit by a sum of exponential functions convoluted with a Gaussian

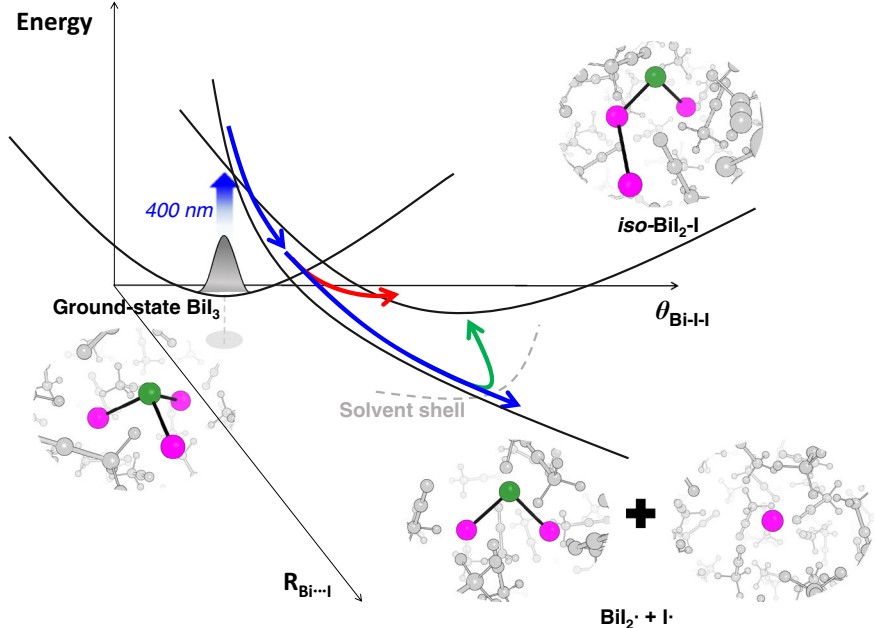

**Fig. 1 A schematic representation for the photodynamics of BiI₃ in acetonitrile.** Upon 400 nm excitation, BiI₃ in acetonitrile solution undergoes isomerization to form *iso*-BiI₂-I and dissociation to form BiI₂• and I• radicals. The kinetics and structural movements at the initial stage of those reactions were investigated in this study. Here, we answer the following three questions using fs-TRXL; (i) roaming-mediated direct isomerization or cage-induced isomerization, (ii) wavepacket trajectory for the isomerization, and (iii) wavepacket trajectory of the dissociation fragments. Qualitative potential energy surfaces are shown along the Bi···I distance ($R_{Bi···I}$) and Bi-I-I angle ($\theta_{Bi-I-I}$). Green and magenta spheres represent the Bi and I atoms, respectively. The solvent cage surrounding the solute molecules is represented by gray spheres. Possible wavepacket trajectories are represented with colored arrows. The blue, red, and green arrows represent the wavepacket trajectories for the dissociation, roaming-mediated direct isomerization, and cage-induced isomerization.

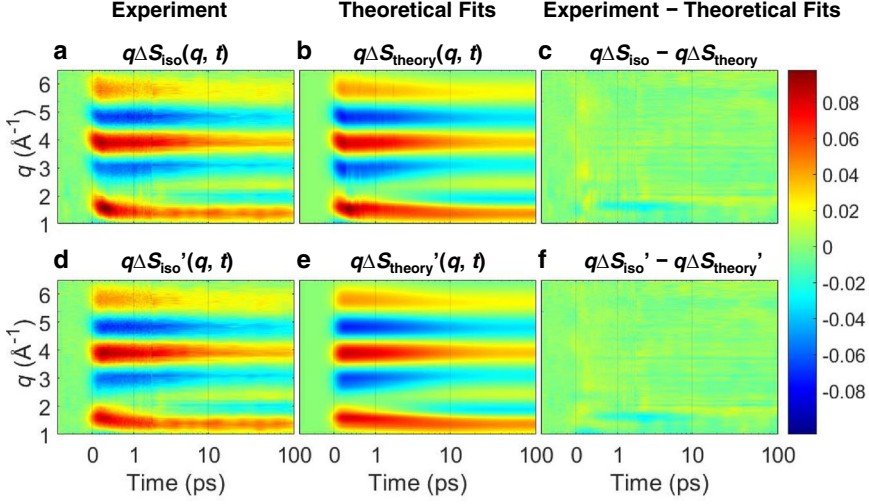

**Fig. 2 Experimentally measured fs-TRXL signals with theoretical fit results. a** Experimental $q\Delta S_{iso}(q, t)$. **b** Corresponding theoretical fit results, $q\Delta S_{theory}(q, t)$, obtained from the global fit combined with structural analysis. **c** Residuals obtained by subtracting theoretical fits from the experimental data. **d** The extracted signals of exponential kinetics, $q\Delta S_{iso}'(q, t)$. **e** Corresponding theoretical fit results, $q\Delta S_{theory}'(q, t)$, obtained from the global fit combined with structural analysis. **f** Residuals obtained by subtracting theoretical fits from the experimental data. All plots share a color scale representing the amplitude of the signal in absolute electronic units per solvent molecule. Source data for panels (**a**) and (**d**) are provided as a Source Data file.

instrumental response function (IRF). As shown in Supplementary Fig. 6a and b, a sum of four exponential functions fit those two RSVs satisfactorily, whereas a sum of three exponential functions did not. The four time constants were determined to be 508 (±13) fs, 3.11 (±0.43) ps, 8.83 (±1.11) ps, and 11.90 (±1.67) ps, and the full width at half-maximum (FWHM) of the IRF was determined to be 162 (±7) fs. First, we analyzed $\Delta S_{iso}'(q, t)$ to obtain the population kinetics and the equilibrium structures of

reacting species, as explained later. Based on these results, the full dataset of the isotropic signals, $\Delta S_{iso}(q, t)$, was analyzed to extract the ultrafast structural motions of the reacting species. The final fitting results presented in Fig. 2b and Supplementary Fig. 7 well describe the experimental features, giving almost zero residual as shown in Fig. 2c. More detailed procedures for the data analysis are given in Methods and Supplementary Methods and will be briefly described in the following sections as well.

The four time constants obtained from the fit of the first two RSVs guided us to build a pool of candidate kinetic models in the search for the best kinetic model. We assumed that three of the four time constants correspond to the solute kinetics and the remaining one to solvent heating dynamics. According to our previous work[37] where the photochemistry of $BiI_3$ upon 400 nm excitation in the time range from 100 ps to 1 µs was investigated, an isomer, $iso$-$BiI_2$-I, and dissociation fragments, $BiI_2$• and I•, are present at 100 ps. However, it was impossible to reproduce the difference scattering signals at earlier time delays, for example at 1 ps, with a linear combination of the known scattering curves of solute and solvent, as shown in Supplementary Fig. 8. This result indicates that at time delays earlier than 100 ps must involve an additional intermediate, termed as "X", whose structure differs from that of the known isomer, $iso$-$BiI_2$-I. We built all possible kinetic models that satisfy these conditions and tested each of them through a global fit analysis (GFA) of $\Delta S_{iso}'(q, t)$. The resultant reduced $\chi^2$ values, $\chi^2_{red}$ (see Supplementary Data 1) obtained from the GFA based on each kinetic model, were compared with each other to determine the best kinetic model. By GFA, the equilibrium structure of the unknown species "X" was determined using the structural parameters for "X" as free parameters for the fit. We note that the optimized structure of "X" was obtained robustly regardless of the kinetic model (see Supplementary Data 2). Detailed methods for determining the best kinetic model are described in Supplementary Methods. The theoretical curves, $q\Delta S_{theory}'(q, t)$, which correspond to the lowest $\chi^2_{red}$ obtained from GFA, are shown in Fig. 2e, giving a good quality of fit, as can be seen in the residual plot in Fig. 2f.

In the best kinetic model, the first time constant, 508 fs, is assigned to the kinetics of the solvent, and the remaining three time constants to the solute kinetics. The best kinetic model and the corresponding time-dependent concentrations of reacting species are shown in Fig. 3a and b, respectively, and the kinetics of bulk solvent temperature is shown in Supplementary Fig. 10. At the early stage of the photoreaction, the excited $BiI_3$ molecules branch out into two reaction pathways, the formation of the X intermediate and dissociation pathways at a branching ratio of about 50:50 (±2). The molecules that fail to completely dissociate to yield a free iodine atom form the X intermediates. Subsequently, 22.58 (±0.05) % of the X intermediate relaxes to $iso$-$BiI_2$-I with the time constant of 8.83 (±1.11) ps, and the other remaining 77.42 (±0.05) % returns to the ground state with the time constants of 3.11 (±0.43) ps. Since the X intermediate is the

precursor of $iso$-$BiI_2$-I and the structure of X corresponds to an isomer, we now name the X intermediate as the early isomer and $iso$-$BiI_2$-I as the late isomer. In the dissociation pathway, the dissociated $BiI_3$ molecules yield $BiI_2$• and I• radicals, and about 60.12 (±0.08) % of those fragments recombine to the ground-state $BiI_3$ with the time constant of 11.90 (±1.67) ps. Considering the time scale of the ground-state recovery from the dissociation fragments[53,54], the recombination of the dissociation fragments to the ground state can be attributed to geminate recombination. The remaining fragments, based on our previous work, non-geminately recombine within hundreds of nanoseconds[37].

In spectroscopic studies on the photodynamics of some halocarbon compounds known to form an isomer species in solution such as $CHI_3$ and $CH_2BrI$, a precursor state termed caged photofragments or hot photofragments or hot isomer state was detected[46,49]. The precursor relaxes to the ground-state isomer with a time constant of 7.0–7.5 ps, which is close to the relaxation time of 8.83 ps observed in this study for the early isomer to transform into the late isomer. Nevertheless, the exact structure of the precursor state has rarely been investigated. Here, we determined the detailed three-dimensional structure of the precursor state, that is, the early isomer of $BiI_3$. The optimized structure of the early isomer determined from the GFA is shown in Fig. 4a and b showing the side view and top view of the structure, respectively, and compared with the structures of ground-state $BiI_3$ and the late isomer, $iso$-$BiI_2$-I. We note that the structure of the early isomer was obtained purely from the free-parameter fitting of the structural parameters, without any help of quantum calculation, during the GFA as detailed in Supplementary Methods. The overall structure of the early isomer is about halfway between the structures of the ground-state $BiI_3$ and the late isomer. Notably, the Bi-$I_b$-$I_c$ angle plays a crucial role in determining the identity of those species. From a structural point of view, the early isomer can switch to either ground-state $BiI_3$ or the late isomer primarily by closing or opening of its Bi-$I_b$-$I_c$ angle, respectively, by ~40°. This structural nature of the early isomer partly rationalizes its kinetics obtained from GFA as well. Furthermore, in contrast to the recent studies, where the radical pair of $CH_2I_2$ was investigated[55,56], the structure of the early isomer was determined without the aid of a Debye-Waller factor (DWF) during the structural refinement in the GFA. In those previous studies, the I⋯I distance of the radical pair of $CH_2I_2$ had to be roughly determined with large mean-squared displacement involved in a DWF or approximated as two dominating distances.

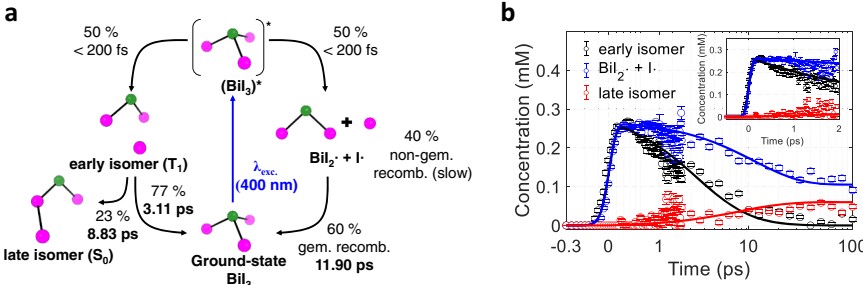

**Fig. 3 Kinetic model, temporal concentration profiles of reacting species of BiI₃. a** Kinetic model for the photochemistry of $BiI_3$ revealed in this work. Upon 400 nm excitation, $BiI_3$ undergoes isomerization to the early isomer and dissociation to radical fragments, $BiI_2$• and I•. The early isomer subsequently relaxes to the late isomer and ground-state $BiI_3$, while a portion of the radical fragments to the ground-state $BiI_3$ through geminate recombination (gem. recomb.). The remaining radical fragments stay almost unreacted within the probed time range due to the slow rate of non-geminate recombination (non-gem. recomb.). **b** The concentration profile of reacting species corresponding to the kinetic model shown in (**a**). The inset shows the early-time concentration profile up to 2 ps in a linear time scale. The solid lines were obtained from the global fit analysis of $\Delta S_{iso}'(q, t)$ based on the kinetic model shown in a and the dots with one-standard-deviation error bars were obtained by independently fitting the individual experimental data with a linear combination of solute, cage, and solvent terms.

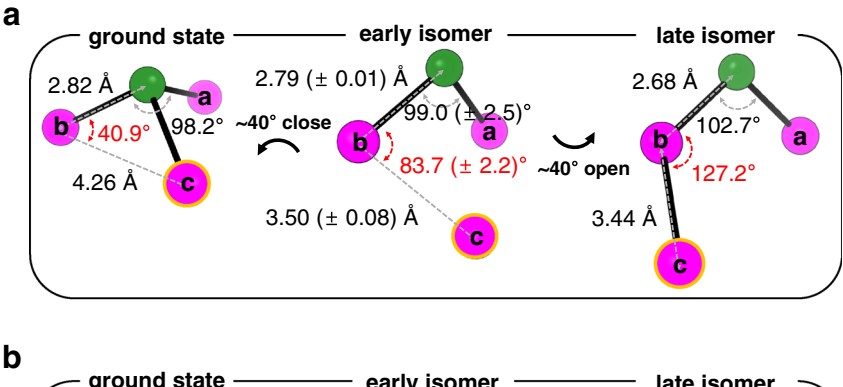

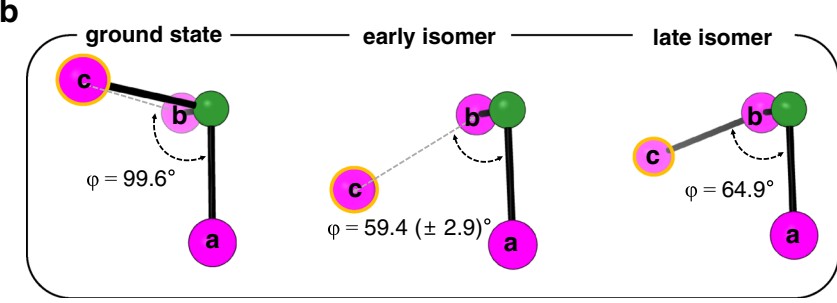

**Fig. 4 The optimized structure of the early isomer. a** Side views and **b** top views of the structure of the early isomer optimized in the global fit analysis (middle) in comparison with the structures of the ground-state $BiI_3$ (left) and late isomer (right). Green and magenta spheres stand for Bi and I atoms, respectively, and three I atoms are labeled with a, b, and c. Among the three I atoms, $I_c$ is highlighted with an orange circle for a more clear comparison of the structures. Side views of the molecules are shown to visualize the overall structures, while top views of the molecules are shown to represent the dihedral angles. The early isomer has the structure that can switch to either the ground-state or the late isomer of $BiI_3$ mainly by closing or opening of its $Bi-I_b-I_c$ angle by ~40°. The error values in parentheses correspond to one standard-deviation of each structural parameter used in the global fit analysis.

Therefore, the newly found structure of the early isomer of $BiI_3$ can be considered an isomeric intermediate with a fairly well-defined structure compared with the precursor isomer of $CH_2I_2$ probably due to much stronger dispersion forces caused by the heavy atoms constituting $BiI_3$.

To support our experimental observation of the early isomer of $BiI_3$, we performed quantum chemical calculations using density functional theory (DFT), and a similar structure to that obtained from the experimental data was found in the triplet state ($T_1$). The DFT-optimized structure of the early isomer is shown in Supplementary Fig. 11 and is compared with the optimized structure from the GFA of the experimental data. The DFT-optimized structure closely reproduces the experimentally observed structural features of the early isomer, especially for the $Bi-I_b-I_c$ angle and $I_b-I_c$ distance. Although the early isomer is assigned to the $T_1$ state following our computational result, it is intrinsically not possible to exclusively determine the electronic nature of the excited state of $BiI_3$ due to the strong spin-orbit coupling (SOC) of Bi and I atoms, which results in a mixing of various singlet and triplet states. For example, the ~400 nm excitation condition already leads to excited states where several singlet and triplet states are mixed (see Supplementary Table 1). Considering the strong SOC in $BiI_3$, it is expected that the intersystem crossing easily occurs, and therefore the early isomer ($T_1$) can effectively relax to the late isomer ($S_0$) or ground-state $BiI_3$ ($S_0$), as shown in Fig. 3a.

**Atomic movements during isomerization and dissociation.** Besides the population kinetics of the reaction intermediates, $\Delta S_{iso}(q, t)$ shows a rapidly varying feature in the sub ~500 fs time range, which was interpreted to arise from atomic movements of reacting species rather than population dynamics between various reacting species. Thus, to obtain the so-called molecular movie based on the wavepacket trajectories, we fit $\Delta S_{iso}(q, t)$ using a set

of structural parameters based on the predetermined kinetic profiles and the equilibrium structures. Briefly, for the $BiI_3$ molecule that undergoes isomerization into the early isomer, a total of five structural parameters were used; one internuclear distance for two Bi-I bonds ($I_a$-Bi and Bi-$I_b$) with the constraint of the two bonds having the same length, and the other four parameters are $Bi-I_b-I_c$ angle, $I_a-Bi-I_b-I_c$ dihedral angle, $I_a-Bi-I_b$ angle, and $I_b\cdots I_c$ distance. For the $BiI_3$ molecule that undergoes a dissociation pathway, three parameters were used; one distance $Bi-I_a$ set to be equal to that of $Bi-I_b$ and the others for $Bi\cdots I_c$ distance along the initial bonding direction and $I_a-Bi-I_b$ angle. During the dissociation, the fragments will rotate and be separated from each other, and the wavepacket representing the dissociating molecule will be quickly dispersed. To compensate for these effects, DWFs were applied to the atomic pairs containing the departing iodine atom, $I_c$, of the dissociating $BiI_3$. More details are described in "Methods" and Supplementary Methods.

The temporal profiles of key structural parameters, (i) $Bi-I_b-I_c$ angle, (ii) $I_a-Bi-I_b-I_c$ dihedral angle of isomerizing $BiI_3$ and (iii) $Bi\cdots I_c$ distance of dissociating $BiI_3$, are shown in Fig. 5a and b, and those of all structural parameters are shown in Supplementary Fig. 12. The initial structural movements for the isomerization are observed through the spreading of $Bi-I_b-I_c$ angle along with the decrease of $I_a-Bi-I_b-I_c$ dihedral angle, as shown in Fig. 5a. This observation indicates that the dissociated $I_c$ atom via the frustrated bond fission of $Bi-I_c$ bond is trapped and roams around the remaining body of $I_a-Bi-I_b$ to form the early isomer, rather than escape the solvent cage. The observed isomerization of $BiI_3$ shows structural movements resembling those of recently reported geminal tri-bromides revealed by semi-classical trajectory computations[15]. In particular, the partially dissociated atom ($I_c$) wanders within the restricted region due to the interaction with the remaining fragment. But, in contrast to those geminal tri-bromides, the wandering motion is mostly predominated by the partially dissociated iodine, rather than the heavier Bi center

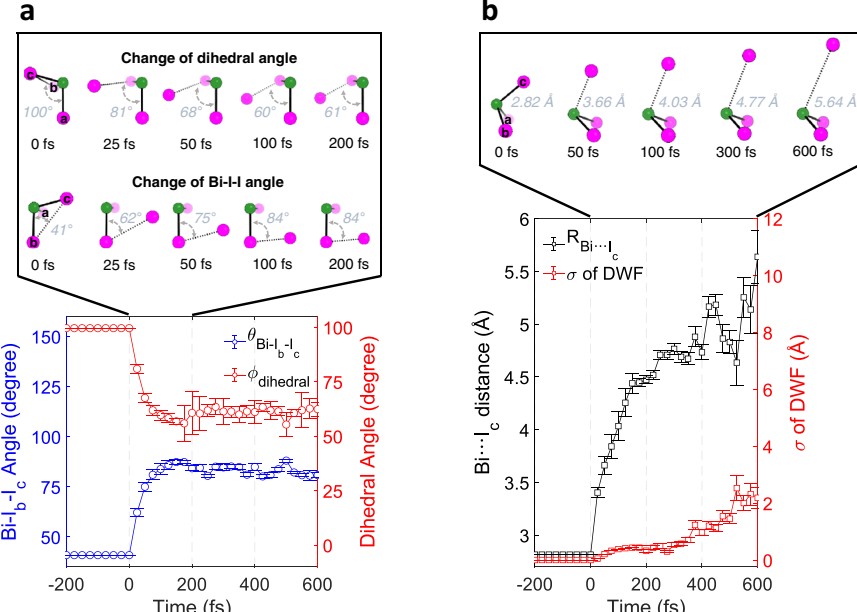

**Fig. 5 Structural parameters selected to represent the structural motion of the initial isomerization, dissociation of BiI₃, and coherent wavepacket. a** The mean Bi-$I_b$-$I_c$ angle and $I_a$-Bi-$I_b$-$I_c$ dihedral angle of the early isomer are shown in blue and red dots, respectively, with the corresponding standard deviations as error bars. **b** The mean Bi···$I_c$ distance of BiI₃ undergoing dissociation and the root-mean-squared displacement ($\sigma$) obtained from the Debye-Waller factor (DWF) are shown in black and red squares, respectively, with the corresponding standard deviations as error bars. The insets in both **a** and **b**, show the snapshots of the molecular structure at some representative time delays. Green and purple spheres stand for Bi and I atoms, respectively, and three I atoms are labeled with a, b, and c. The structural parameters shown in (**a**) and (**b**) were obtained from the structural fit analysis on $\Delta S_{iso}(q, t)$. The error bars in (**a**) and (**b**) at time delays earlier than 175 fs are one-standard-deviation error values calculated from the covariance matrix of the coefficients of quartic polynomials used to model the parameters during the structural fit analysis. The error bars of structural parameters in a and b at time delays later than 175 fs are the averaged one-standard-deviation error values obtained from the fits starting from ~5000 random structures for each time delay. The error bars for the σ of the DWF at time delays later than 175 fs are the one-standard-deviation error values obtained from the fit performed at each time delay.

atom. According to our result shown in Fig. 5a, these motions occur in ~100 fs, which is much faster than the time scale of the geminate recombination via a single collision with the solvent cage, which has been estimated to be around 250–500 fs[57]. Therefore, we can infer that the observed ultrafast isomerization is completed without any aid of a solvent cage.

We thus attribute this isomerization accomplished via the wandering motion of the incipient iodine fragment in the long-range distance (3–4 Å) to roaming-mediated direct isomerization rather than cage-induced isomerization. The ultrafast time scale of the isomerization suggests that there might be a conical intersection through which the initial nuclear movements for the isomerization can effectively take place non-adiabatically[15,58]. We note that introducing DWFs to the atomic pairs containing the roaming iodine atom ($I_c$) gives the σ value of around 0.1 Å, which is comparable to the structural uncertainties of the distances for those atomic pairs. Therefore, one possibility is that only the changes of the mean structure of BiI₃ along the net direction of the roaming reaction are observed with the current TRXL experiment whereas other stochastic movements involved in the roaming reaction are not resolved within the structural error range and temporal resolution of our signal. Regarding the obtained σ values, more details are given in Supplementary Note 1. Even though the temporal and spatial resolutions of this study do not allow tracking the full wavepacket trajectories, our results successfully visualized the time-resolved structural motion related to the roaming mechanism in solution.

The structural motion for dissociation was captured in the form of the increasing distance between the bismuth atom and one of the three iodine atoms, as shown in Fig. 5b. It took <150 fs

to reach a distance of about 150 % of the initial bond length, where the chemical bond can be considered broken. The Bi···$I_c$ distance was tracked up to ~5.6 Å, beyond which the wavepacket representing the dissociating molecule was too dispersed to be captured by the current structural sensitivity. The increasing σ of the DWF reflects this broadening of the wavepacket of the dissociating molecule. The dissociation was initiated by a fast elongation of Bi···$I_c$ distance up to ~125 fs with the average velocity of ~0.011 Å fs$^{-1}$ and, subsequently, the dissociated fragments move further away from each other to reach the observed maximum Bi···$I_c$ distance of ~5.6 Å with a slower average velocity of ~0.003 Å fs$^{-1}$. The slowing down of the departing rate of the wavepacket indirectly reflects the loss of kinetic energy due to the collision with the solvent shell. In addition, the TRXL data exhibits a coherent vibrational wavepacket motion. As shown in Supplementary Fig. 13a, an oscillatory signal can be seen in the first RSV, and the main frequency of this oscillation was found to be ~45 cm$^{-1}$. Among the temporal profiles of the structural parameters shown in Supplementary Fig. 12, this oscillatory component was identified in the temporal change of $I_a$-Bi-$I_b$ angle of dissociating BiI₃, as shown in Supplementary Fig. 13b, together with the oscillatory signal obtained by subtracting the exponential component from the first RSV. This result indicates that the observed coherent oscillation corresponds to the bending motion of BiI₂ fragment. The vibrational frequencies obtained from the DFT calculation support our experimental observation. As summarized in Supplementary Fig. 14, the $I_a$-Bi-$I_b$ bending motion with a frequency of 52 cm$^{-1}$, which is similar to the experimentally observed frequency of 45 cm$^{-1}$, is indeed identified.

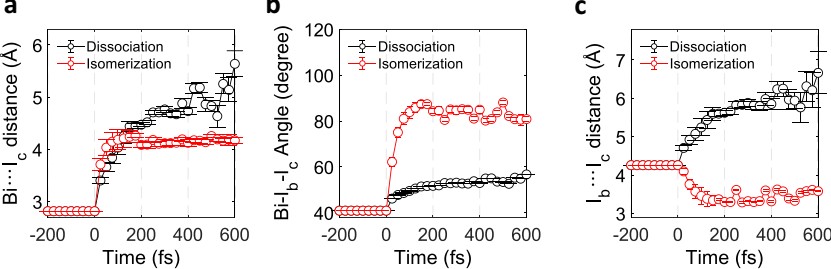

**Fig. 6 Comparison of the structural motions for the dissociation and roaming-mediated isomerization channels. a** Bi···$I_c$ distance, **b** Bi-$I_b$-$I_c$ angle, and **c** $I_b$···$I_c$ distance for the $BiI_3$ in the dissociation pathway (black) and the roaming-mediated isomerization pathway (red) are shown. The structural motions are obtained in the structural fit on $\Delta S_{iso}(q, t)$. The error bars represent the one-standard-deviation error values estimated from the relevant error values obtained from the covariance matrix of the coefficients of quartic polynomials for the time delays earlier than 175 fs and the averaged one-standard-deviation error values obtained from the fits starting from ~5000 random structures for the time delays later than 175 fs.

Finally, comparisons of the relevant structural parameters, the Bi···$I_c$ distance, the $I_b$···$I_c$ distance, and the Bi-$I_b$-$I_c$ angle, for the dissociation pathway and the roaming-mediated isomer-formation pathway vividly visualize that different types of atomic motions are involved in the two reaction pathways (Fig. 6). For both pathways, the Bi···$I_c$ distance similarly increases up to ~100 fs (Fig. 6a). Then, the Bi···$I_c$ distance stops increasing for the isomer-formation pathway, demonstrating the nature of the frustrated fission of the Bi···$I_c$ bond for the roaming-mediated reaction whereas it keeps increasing for the dissociation pathway. In contrast, the Bi-$I_b$-$I_c$ angle shows a significant increase within ~100 fs and stays constant afterward for the isomer formation whereas its increase is relatively marginal throughout the whole time scale for the dissociation pathway (Fig. 6b). The contrast between dissociation and the roaming-mediated isomer formation is even clearer in the time profile of $I_b$···$I_c$ distance, which keeps increasing for dissociation and initially decreases up to ~100 fs for isomer formation (Fig. 6c).

In summary, we have elucidated both the kinetics and the structural changes of reacting molecules involved in a roaming reaction up to 100 ps from the beginning of the photoreaction of $BiI_3$ in solution (Supplementary Fig. 15). Specifically, the wavepacket trajectories along the isomerization and dissociation pathways were tracked via taking snapshots of the structures of reacting molecules. We highlight that the early isomer is formed through a roaming-like pathway within 100 fs and the characteristic structural motions of the wandering iodine radical fragment are traced through the ultrafast changes in the Bi-$I_b$-$I_c$ angle, $I_a$-Bi-$I_b$-$I_c$ dihedral angle, the Bi···$I_c$ distance, and the $I_b$···$I_c$ distance. This work demonstrates the feasibility of experimentally visualizing the very details of atomic movements involved in a roaming reaction in solution using fs-TRXL.

## Methods

**Femtosecond time-resolved X-ray liquidography (fs-TRXL) experiment**. The experiment was conducted at the XSS beamline of PAL-XFEL (the Pohang Accelerator Laboratory X-ray Free-Electron Laser)[59,60] with the pump-probe setup. The XEFL delivered X-ray pulses with an energy of 12.7 keV and a temporal width <50 fs at a repetition rate of 30 Hz. The X-ray beam was focused to a spot of ~30 μm diameter. Optical pump pulses at the wavelength of 400 nm were used and spatially focused to a spot of ~200 μm diameter, yielding a laser fluence of ~1.8 mJ mm$^{-2}$. The sample solution of $BiI_3$ in acetonitrile at a concentration of 1 mM was used for the fs-TRXL experiment, and 0.5 mM solution of a dye, 4-bromo-4′-(N,N-diethylamino)-azobenzene (HANCHEM, 99.9%), in acetonitrile was used to measure the solvent heating signals. The scattering images were measured at 134 pump-probe time delays, which were −950 fs–1.95 ps (with 25 fs time interval), 2.51, 3.16, 3.98, 5.01, 6.31, 7.94, 10, 12.5, 15.8, 19.9, 25.1, 31.6, 39.8, 501, 63.0, 79.4, and 100 ps. The laser-off images taken at a time delay of −3.0 ps were obtained repeatedly before taking every laser-on image. More details are described in Supplementary Methods.

**Data processing**. Time-resolved two-dimensional (2D) difference scattering images were obtained by subtracting the laser-off images from the laser-on images. The 2D difference scattering images were converted to one-dimensional (1D) azimuthally averaged difference scattering curves, $\Delta S_{azimut}(q, t)$, by calculating the average intensity as a function of the momentum transfer, $q = (4\pi\lambda^{-1})\cdot\sin(2\theta\cdot2^{-1}) = (4\pi\lambda^{-1})\sin[1/2\cdot\tan^{-1}(l\cdot d^{-1})]$, where $\lambda$ is the wavelength of x-ray, the $2\theta$ is the scattering angle, $l$ is the distance from the beam center to a given pixel and $d$ is the sample-to-detector distance (see Supplementary Fig. 2). The resultant scattering curves are contaminated by systematic noise. To get rid of the systematic noise, we applied the SANOD method[61] to $\Delta S_{azimut}(q, t)$ and used the output $\Delta S_{azimut}(q, t)$ for the subsequent data processing. More details are described in Supplementary Methods and are summarized in Supplementary Fig. 4.

**Extraction and SVD of $\Delta S_{iso}(q, t)$ and $\Delta S_{aniso}(q, t)$**. As can be seen in Supplementary Fig. 2, the raw difference scattering images are anisotropic, and therefore $\Delta S_{azimut}(q, t)$ can be decomposed into anisotropic and isotropic components. Based on a well-established method[51,52], which is described in Supplementary Methods, the 2D difference scattering images at each time delay were decomposed into 1D anisotropic and isotropic difference scattering curves, $\Delta S_{aniso}(q, t)$ and $\Delta S_{iso}(q, t)$, respectively (see Supplementary Fig. 4). To examine the temporal behavior of the scattering data, we applied the SVD analysis to both $\Delta S_{aniso}(q, t)$ and $\Delta S_{iso}(q, t)$. A detailed explanation of SVD is described in Supplementary Methods.

The SVD of $\Delta S_{iso}(q, t)$ yielded four significant RSVs (see Supplementary Fig. 5). The first five RSVs of $\Delta S_{iso}(q, t)$ of the $BiI_3$ solution weighted by the corresponding singular values are shown in Supplementary Fig. 5a. Among those RSVs, third and fourth RSVs ($RSV_3$ and $RSV_4$) mainly contribute to the signal on the sub-ps time scales, and when the two RSVs are removed from $\Delta S_{iso}(q, t)$, the ultrafast shifts of the positive and negative peaks in $q$-space occurring at <500 fs, which are expected to arise from the coherent structural motions of $BiI_3$, disappear, as shown in Supplementary Fig. 17. Therefore, we can infer that the first two RSVs of $\Delta S_{iso}(q, t)$ of the $BiI_3$ solution contain the population kinetics of reacting chemical species while $RSV_3$ and $RSV_4$ are associated with coherent motions of $BiI_3$.

In Supplementary Fig. 3, the first two singular vectors (SVs) of $\Delta S_{aniso}(q, t)$ are shown in comparison with $\Delta S_{aniso}(q, t)$ of a dye solution, and it can be seen that $\Delta S_{aniso}(q, t)$ are dominated by two singular values. We note that, as shown in Supplementary Fig. 3c, d, the first two SVs obtained from $\Delta S_{aniso}(q, t)$ of $BiI_3$ solution are nearly identical to those obtained from $\Delta S_{aniso}(q, t)$ of a dye solution. Since $\Delta S_{aniso}(q, t)$ of the dye solution originates from the optical Kerr effect (OKE) of neat solvent, that is, transient alignment of solvent molecules induced by polarized optical field, the similarity of $\Delta S_{aniso}(q, t)$'s of the $BiI_3$ and dye solutions indicates that the structural dynamics of $BiI_3$, which is of our interest, are not reflected in $\Delta S_{aniso}(q, t)$ but contained only in $\Delta S_{iso}(q, t)$. Therefore, only $\Delta S_{iso}(q, t)$ of the $BiI_3$ solution was further analyzed.

**Extraction of the difference scattering signal, $\Delta S_{iso}'(q, t)$, representing the exponential kinetics**. The $\Delta S_{iso}(q, t)$ of the $BiI_3$ solution has rich features including the signals from both exponential kinetics and coherent dynamics. To simplify the analysis, we first extracted and analyzed the contribution corresponding to the exponential kinetics, termed $\Delta S_{iso}'(q, t)$. To extract the exponential kinetics, the first two RSVs of $\Delta S_{iso}(q, t)$, which are responsible for the exponential kinetics, were globally fitted using a sum of exponential functions convoluted with a Gaussian IRF. As shown in Supplementary Fig. 6a, b, a sum of four exponential functions fit those RSVs satisfactorily, whereas a sum of three exponential functions did not. The fit using a sum of four exponential functions on the first two RSVs results in four time constants, 508 (±13) fs, 3.11 (±0.43) ps, 8.83 (±1.11) ps, and 11.90 (±1.67) ps, and the FWHM of the IRF was determined to be 162 (±7) fs. To reconstruct $\Delta S_{iso}'(q, t)$, the first and second RSVs in the V matrix of $\Delta S_{iso}(q, t)$ were replaced by their corresponding fit curves, and the third and fourth SVs were

filtered out by setting the corresponding singular values, LSVs and RSVs to zero in the S, U and V matrices, generating a new set of matrices, U', S', and V'. Using the relation $M = U \cdot S \cdot V^T$, $\Delta S_{iso}'(q, t)$ were constructed in the form of $U' \cdot S' \cdot V'^T$, where the superscript T means the transpose of a matrix. These procedures are relevant only for the scattering data at early time delays, where coherent dynamics are present, and therefore were applied to the data up to the time delay of 2 ps, from which $\Delta S_{iso}'(q, t)$ becomes almost identical to $\Delta S_{iso}(q, t)$. Since the coherent dynamics are nearly absent after 2 ps in our data, $\Delta S_{iso}'(q, t)$ was set to be equal to the original data, $\Delta S_{iso}(q, t)$, after 2 ps. The reconstructed signals, $\Delta S_{iso}'(q, t)$, representing the exponential kinetics, are shown in Fig. 2d and Supplementary Fig. 18. More details of theoretical rationales for this procedure are described in Supplementary Methods.

**GFA on $\Delta S_{iso}'(q, t)$.** To analyze $\Delta S_{iso}'(q, t)$, we built the theoretical curves, $\Delta S_{theory}'(q, t)$ based on kinetic models as well as our prior knowledge from the previous TRXL study of $BiI_3$ solution with a 100 ps temporal resolution. Details are described in Supplementary Methods. Briefly, based on the four time constants obtained from the SVD analysis on $\Delta S_{iso}'(q, t)$, we devised eleven possible kinetic model frames for solute molecules and the temperature rise of the bulk solvent was modeled via a single exponential function. By distributing three of the four time constants to one of the eleven kinetic model frames for the solute molecules and assigning the remaining time constant to the solvent kinetics, we built a total of 264 possible kinetic models. In those kinetic models, a new intermediate (the early isomer) was introduced because the experimental signals cannot be explained only with the known species from the previous TRXL study of $BiI_3$ solution, which are the late isomer ($iso$-$BiI_2$-I) and dissociation fragments ($BiI_2\bullet$ and $I\bullet$). Each of the kinetic models was tested through GFA on $\Delta S_{iso}'(q, t)$ using theoretical curves, $\Delta S_{theory}'(q, t)$. Details for constructing theoretical curves, $\Delta S_{theory}'(q, t)$, are described in the Supplementary Information. In GFA, $\Delta S_{iso}'(q, t)$ were globally fit against the $\Delta S_{theory}'(q, t)$ by minimizing the reduced $\chi^2$ ($\chi^2_{red}$) value, which is defined as follows

$$\chi^2_{red} = \frac{1}{N - p - 1} \sum_{j=\text{time delay}} \sum_i \frac{(\Delta S_{theory}'(q_i, t_j) - \Delta S_{iso}'(q_i, t_j))^2}{\sigma^2_{i,j}} \quad (1)$$

where $N$ is the total number of data points along the $q$- and $t$-axes, $p$ is the number of fit parameters, and $\sigma_{i,j}$ is the standard deviation of the difference scattering intensity at $i^{th}$ $q$ of $j^{th}$ time delay. The minimization of the $\chi^2_{red}$ was performed with the MINUIT software package and the error values were obtained with the MINOS algorithm in MINUIT[62]. In GFA, the structure of the early isomer and the branching ratio between the reacting species were optimized. We note that, during GFA, structures for the late isomer ($iso$-$BiI_2$-I) and dissociation fragments ($BiI_2\bullet$ and $I\bullet$) were fixed to the structures reported in the previous TRXL study of $BiI_3$ solution. Details for the fitting parameters in GFA are given in Supplementary Methods. In Supplementary Data 1, we summarize the relative $\chi^2_{red}$ values with respect to the minimum value obtained from GFA on the total 264 kinetic models. The best kinetic model and corresponding concentration profiles obtained from GFA are shown in Fig. 3a and b. The optimized equilibrium structure of the early isomer from GFA based on the best kinetic model is shown in Fig. 4.

**Structural analysis on $\Delta S_{iso}(q, t \geq 175\,fs)$ and $\Delta S_{iso}(q, t < 175\,fs)$.** Based on the kinetics and the equilibrium structures of the intermediates determined in the GFA on $\Delta S_{iso}'(q, t)$, the coherent structural motions for the reacting molecules were obtained by fitting the $\Delta S_{iso}(q, t)$. Specifically, to get the snapshots of the molecular structures, each time delay of the $\Delta S_{iso}(q, t)$ at the time delays were fitted with $\Delta S_{theory}(q, t)$ obtained by varying the structures of $BiI_3$ in action along the isomerization and dissociation reaction pathways. For analyzing the experimental signals around $t = 0$, however, the temporal broadening of the signals by the IRF should be considered. Otherwise, the direct structural fit on the signals leads to a wrong result as described in Supplementary Note 2.

Thus, for $\Delta S_{iso}(q, t)$ at time delays smaller than the FWHM of the experimental IRF, that is $\Delta S_{iso}(q, t < 175\,fs)$, each of the time-dependent structural parameters was modeled by a quartic polynomial function as follows

$$x(t) = \sum_{k=0}^{4} a_{4-k} t^{4-k} \quad (2)$$

where $a_{4-k}$ is the coefficient of the polynomial function. Then the constructed instantaneous theoretical difference scattering curves, $\Delta S_{inst}(q, t)$, were convoluted with the Gaussian IRF resulting in $\Delta S_{theory}(q, t < 175\,fs)$. For the structural fit on $\Delta S_{iso}(q, t < 175\,fs)$, $\chi^2_{red}$ is defined as follows

$$\chi^2_{red} = \frac{1}{N - p - 1} \sum_{j=\text{time delay}} \sum_i \frac{(\Delta S_{theory}(q_i, t_j) - \Delta S_{iso}(q_i, t_j))^2}{\sigma^2_{i,j}} \quad (3)$$

where $N$ is the total number of data points along the $q$- and $t$-axes, $p$ is the number of fit parameters, and $\sigma_{i,j}$ is the standard deviation of the difference scattering intensity at $i^{th}$ $q$ of $j^{th}$ time delay. Furthermore, to suppress possible artifact caused by use of a polynomial function, we applied penalties to parameters that inverted

the sign of the first derivative. The penalty term ($P$) is defined as follows

$$P = \alpha \cdot \sum_{i \in C} \left( \left| \sum_j \left( \frac{(\partial x_i / \partial t)_{t=t_j}}{\max[(\partial x_i / \partial t)]} \right) \right|^{-1} \right) \quad (4)$$

where $\alpha$, which is set to be 100, is a constant weighting factor, $x_i$ is the $i$-th structural parameter belonging to the subset $C$ in which the sign of the first derivative of the parameter changes, $t_j$ is the $j$-th time delay, $(\partial x_i / \partial t)_{t=t_j}$ is the value at $t = t_j$ of the first derivative of $x_i$ with respect to $t$, and $\max[(\partial x_i / \partial t)]$ is the maximum value of the first derivative of $x_i$ with respect to $t$. The coefficients of the polynomial functions for each structural parameter were optimized by minimizing a sum of $\chi^2_{red}$ and $P$, which are defined in Eqs. (3) and (4), respectively. More detailed procedures for modeling and constructing the $\Delta S_{inst}(q, t)$ and $\Delta S_{theory}(q, t < 175\,fs)$ are described in Supplementary Methods. The standard deviations for the parameters were calculated based on the covariance matrix, which was calculated during the fit of the coefficients of quartic polynomials using the following equation

$$\sigma^2_{x(t)} = \sqrt{\sum_{k=0}^{4} (t^{4-k})^2 \text{Var}(a_{4-k}) + 2 \sum_{i=0}^{4} \sum_{j=i+1}^{4} t^{4-i} t^{4-j} \text{Cov}(a_{4-i}, a_{4-j})} \quad (5)$$

where $\sigma_{x(t)}$ is the standard deviation of the parameter $x$ at time delay $t$, $\text{Var}(a_{4-k})$ is the variance of the coefficient $a_{4-k}$ and $\text{Cov}(a_{4-i}, a_{4-j})$ is the covariance between the coefficients $a_{4-i}$ and $a_{4-j}$.

For $\Delta S_{iso}(q, t)$ at time delays larger than the FWHM of the experimental IRF, that is $\Delta S_{iso}(q, t \geq 175\,fs)$, the fit was performed at each time delay independently using the following equation

$$\chi^2_{red}(t \geq 175\,fs) = \frac{1}{N_q - p - 1} \sum_i \frac{(\Delta S_{theory}(q_i, t) - \Delta S_{iso}(q_i, t))^2}{\sigma(q_i, t)^2} \quad (6)$$

where $N_q$ is the number of $q$ points, $p$ is the number of parameters, $\sigma(q_i, t)$ is the standard deviation of the $\Delta S_{iso}$ at $q_i$ of time delay $t$. For each time delay, a random structural pool consisting of ~5000 structures was generated and those structures were optimized during the fit and the corresponding $\chi^2_{red}$ was obtained. With respect to the minimum value of $\chi^2_{red}$, $\min[\chi^2_{red}(t)]$, at the given time delay among the obtained $\chi^2_{red}$ ($t$), the output structures giving <1.00001 of the $\min[\chi^2_{red}(t)]$ were collected and the parameters as well as the corresponding error values of those structures were averaged. Detailed parameter conditions and methods for constructing $\Delta S_{theory}(q, t \geq 175\,fs)$ are described in Supplementary Methods.

Finally, the resultant $\Delta S_{theory}(q, t < 175\,fs)$ was concatenated with $\Delta S_{theory}(q, t \geq 175\,fs)$, giving rise to $\Delta S_{theory}$ as shown in Fig. 2b, Supplementary Fig. 7 and Supplementary Fig. 18. The optimized structural parameters were concatenated with the corresponding parameters which were used to calculate the solute signal of $\Delta S_{theory}(q, t \geq 175\,fs)$ and are represented in Fig. 5 and Supplementary Fig. 12.

## Data availability
The time-resolved isotropic difference scattering data analyzed in this study corresponding to Fig. 2a and d are provided as the Source Data file. All relevant data that support the findings of this study are available from the corresponding author upon reasonable request. Source data are provided with this paper.

## Code availability
The codes used for the analysis here are available from the corresponding author on reasonable request.

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

## Acknowledgements

This work was supported by the Institute for Basic Science (IBS-R004). The experiments were performed at the XSS of PAL-XFEL (proposal no. 2020-2nd-XSS-001 and 2019-1st-XSS-011). The authors thank Doo-Sik Ahn, Chi Woo Ahn, Jae Hyuk Lee, Jaeku Park, Intae Eom, and Sae Hwan Chun for their dedicated support for the beamtime experiments.

## Author contributions

H.I. supervised the project; H.I and E.H.C. designed the experiment; E.H.C., J.G.K., Ju. Kim and H.I. developed the data analysis strategy; E.H.C., Ju. Kim, H.K., Y.L., S.L., Je. Kim and H.I. performed the experiments; E.H.C. analyzed the data; K.Y. and Jo. Kim performed quantum chemical calculations; E.H.C., J.G.K., Jo. Kim and H.I. wrote the manuscript with contributions from all authors.

## Competing interests

The authors declare no competing interests.
