## [Peer Review File · Nature Communications]

REVIEWER COMMENTS

Reviewer #1 (Remarks to the Author):

The article Filming ultrafast roaming-mediated isomerization of bismuth triiodide in solution by Choi et al. presents ultrafast x-ray scattering investigation of bond isomerization. This study follows the Nature Chemistry study of Tarnovsky and co-workers, where they combined ultrafast optical spectroscopy and quantum chemical calculations to conclude bond isomerization in tri-bromide systems (BBr_3 , PBr_3 , and CHBr_3) in solution occur via a roaming mechanism. The direct structural resolution of the experiments presented by Choi et al. represent an important advance compared to the study and Tarnovsky, but I have significant questions about the analysis and modeling of the signal that must be clarified before I will be able to recommend the manuscript for publication.

The first area that must be strengthened and clarified is the definition of a roaming reaction, what the clear signatures of such reactions are, and how they would manifest in the x-ray scattering measurement. In my role as reviewer, I have investigated the literature on roaming reactions and find the description in the manuscript inadequate. Specifically, the definition of roaming as a, "Reaction yielding products via reorientational motion in the long-range region of the potential," is incomplete. My understanding is that roaming involves unimolecular reactions where reactants and products are not separated by a narrowly-defined (specific molecular geometry) transition state, but rather a reaction where roughly flat regions of the free energy surface are critical to the reaction mechanism. For the study of Tarnovsky and co-workers, they claimed via simulation that wandering of the light component of the molecule (CH for CHBr_3 in Fig. 3 Nature Chem. 7, 562 (2015)) is central to the roaming reaction mechanism, in addition to the migration of a Br atom to form the new Br-Br bond of the isomer. This brings up two questions regarding the BiI_3 investigated in this study. First, small and lower mass fragment have been central to the other roaming reactions investigated, but this is not consistent with BiI_3 . Which molecular fragment would be roaming/wandering for BiI_3 ? Secondly, such a reaction mechanism should lead to significant increases in the distribution of bond lengths during the roaming reaction prior to bond isomer formation, but no evidence of increased distributions (whether it be a time dependent Debye-Waller or more sophisticated presentation of the distribution) is discussed. The issue of the distribution in bond lengths is addressed for the dissociation channel, but not the roaming reaction.

The second area that must be strengthened is the justification for the structural and kinetic models in the manuscript. Regarding the structural modeling, the isotropic difference scattering signal measures changes in the pair distribution function and is not specifically sensitive to bond angles or the three dimensional geometry of the solute. (As an aside, the anisotropic difference scattering is sensitive to bond angles, but this signal has not been analyzed in the manuscript. An explanation for why the authors chose not to analyze the anisotropic signal would be welcomed by this reviewer.) This means the x-ray scattering measurement in isolation underdetermines the three dimensional structure because there will be many ways of arranging the four atoms of BiI_3 that give the same pair distribution function, but different bond angles. The manuscript needs to more clearly explain how the quantum chemistry structures are using in the structure refinement and how this constrains the search space for the data analysis.

Regarding the kinetic modeling of the experimental findings, I have questions about the uniqueness and robustness of the analysis and require better evidence that the modeling and analysis is consistent with the claim a roaming reaction is occurring. First the SVD of the data and the structural modeling. The legitimacy of treating the time dependent signal as a sum of right singular vectors that can be associated with specific intermediates or products requires the geometric structure of those intermediates/products to be time independent. The clear shifts in the scattering signal during the first picosecond would indicate that any kinetic model would be dubious for the first picosecond. I also

question the need for four exponential time constants in the measurement. Including the error bars, the 8.8 ps and 11.9 ps time constants nearly overlap, particularly considering the fit shown in Fig. S13 appears to have an offset. Lastly and most importantly, a model for a roaming reaction that involves the time dependent rise in an intermediate isomer with a fixed structure does not possess the essential dynamics of a roaming reaction. Where is the roaming/wandering? I fully appreciate the significant challenges associated with such modeling, but without it where is the evidence in support of the primary conclusions from the manuscript?

In summary, the author have designed an interesting study of bond photoisomerization in solution that would put the importance of roaming reactions in solution on firmer experimental ground, but at present the details of the analysis due not prove convincing.

Reviewer #2 (Remarks to the Author):

This is a study on the dissociation and isomerization dynamics and kinetics of bismuth triiodide (BiI₃) in acetonitrile solution. The authors used time-resolved X-ray scattering with femtosecond temporal resolution to measure the intramolecular structural rearrangements of the molecule upon optical excitation. They find a roaming-mediated isomerization process via a hitherto undetected transient intermediate ("early isomer" in a triplet state) and they relate this interesting process to parallel pathways (dissociation and geminate recombination) and to pathways that the same authors or part of the authors revealed in an earlier study at a synchrotron radiation source on slower time scales.

I find this a very convincing study and I think that the data and presentation are just wonderful. The manuscript is very clear, the results are exceptionally clear-cut and the discussion is very focused and of outstanding accessibility.

While solvent cage-induced isomerization in similar molecules is thought to proceed on time scales of several hundreds of femtoseconds and beyond, where collisions with solvent molecules induce radical recombination, here the authors clearly detect a new mechanism. Within about 100 fs they find the direct formation of an "early isomer", a time scale that is clearly much shorter than typical time scales of solute-solvent interactions. They very convincingly show how this isomer results from (coherent) nuclear motions directly induced by the electronic excitation and how it then proceeds to the final isomer (or back to the initial ground state in a parallel pathway). The authors identify a particular I atom in the system that does not quite dissociate ("frustrated bond fission") and instead "wanders around", trapped in the solvent cage in the vicinity of the remaining fragment. At the same time, this "wandering around" happens so fast (in about 100 fs) that it is too fast for the solvent cage to participate or even determine the motions of the solute atoms. A fascinating case, where comparably long-range interactions within an "almost dissociated molecule" in solution (at distances of 3-4 Å) lead to very interesting molecular transformations.

I find the level of detail to which the authors followed nuclear motions in time and to which they characterized the structure of the essential reaction intermediate (Figures 3+4) very fascinating. The system, consisting of comparably heavy atoms for good X-ray scattering signals, is well chosen and wonderfully serves the purpose of distinguishing numerous parallel reaction pathways and identifying the one that is really interesting, which the authors rightfully term "roaming-mediated isomerization". This is an excellent demonstration of the capabilities of the employed technique and it is a very convincing demonstration of the importance of "roaming reactions" in solution, a topic that has increasingly gained attention in recent years. It will be very interesting to see how such measurements can be used in the future to see even finer details such as, maybe, how solute-solvent

interactions do interfere with the observed intramolecular roaming process by manipulating the potential energy landscape. Certainly, a great step towards understanding photochemical transformation on the atomic length and time scales.

I therefore believe without any doubt that this manuscript can be published in Nature Communications as is. With the attempt to maybe help improving the one or the other aspect, I suggest that the authors consider my detailed comments below.

Introduction, p. 3 "indicating that roaming reaction is working in the liquid solution phase as well". Given the text-book character of the work which should be very accessible to graduate students as well, I wonder whether the authors could add 1-2 sentences on why (they think that) roaming reactions in solution are thought to be unusual. Is that because they are hard to observe or are there conceptual arguments? Later in the same paragraph they mention a "flat region of the S1" potential energy surface and maybe this is a conceptual framework in which what I consider missing could be formulated.

P. 5, "As a result, we obtained four time constants..." Understanding that this may be hard or impossible (?), can the four time scales, or some of them, even if indirect or qualitative, be related to (some of) the experimental observables in Figure 2? A few sentences here could help to better relate the structural and kinetic information with what was actually measured.

Figure 1: I understand that this is a figure that is supposed to motivate the present study instead of summarizing the inferred reactions pathways. Still, maybe the figure and its content can be enhanced by adding time scales that are known from other studies (such as their previous study referenced in ref. 18)? Also, I have a hard time "seeing" the first process/scenario in the blue inset ("isomerization via a bimolecular recombination"). Maybe it is possible to include both, this and the other scenario of a roaming mediated reaction in the figure to further enhance how meaningful and far-reaching the detection of the roaming mechanism is? Would, e.g., bimolecular recombination happen via a new pathway that would connect the depicted pathways along R_Bi...I and along Theta_Bi-I-I?

Figure 3: Can, in panel (a), the time scales for the very first steps (50% and 50%) be indicated (even if these may happen within the temporal resolution of the experiment)? From the four time constants mentioned in the text (see above), the one with 500 fs is missing in this figure and it may be instructive to add it to the figure caption with a short note that it relates to solvent heating instead of solute intramolecular structural rearrangements. In panel (c), can the I atoms be numbered 1-3 in the top and bottom panels to better "see" the different views of a given structure (up/down) and to better follow the very nice transformations of the molecules (left/right)?

Reviewer #3 (Remarks to the Author):

The paper is an interesting and convincing study of a simple isomerization reaction in solution. So it is an important step forward in this active field and I recommend publication.

I do think a few relevant papers need to be cited.

Photodissociation dynamics of nitromethane and methyl nitrite by infrared multiphoton dissociation imaging with quasiclassical trajectory calculations: Signatures of the roaming pathway, Dey et al. J. Chem. Phys. 140, 054305 (2014);

Ultrafast Proton Transfer Dynamics on the Repulsive Potential of the Ethanol Dication: Roaming-Mediated Isomerization versus Coulomb Explosion, Wang et al J. Phys. Chem. A 2020, 124, 14, 2785

Capturing roaming molecular fragments in real time, Endo et al, Science 370, 1072 (2020)

Also some real time roaming papers from the Dantus group.

Figure 1 is indeed highly schematic and unfortunately misleading. Real wavepackets on anharmonic, reactive potentials do not maintain simple Gaussian shapes; quite the opposite, e.g., major breakup into packet pieces. So I strongly suggest dropping those imagined wavepackets and just use arrows.

Responses to the comments from Reviewer #1

*The article *Filming ultrafast roaming-mediated isomerization of bismuth triiodide in solution* by Choi et al. presents ultrafast x-ray scattering investigation of bond isomerization. This study follows the Nature Chemistry study of Tarnovsky and co-workers, where they combined ultrafast optical spectroscopy and quantum chemical calculations to conclude bond isomerization in tri-bromide systems (BBr₃, PBr₃, and CHBr₃) in solution occur via a roaming mechanism. The direct structural resolution of the experiments presented by Choi et al. represent an important advance compared to the study and Tarnovsky, but I have significant questions about the analysis and modeling of the signal that must be clarified before I will be able to recommend the manuscript for publication.*

The first area that must be strengthened and clarified is the definition of a roaming reaction, what the clear signatures of such reactions are, and how they would manifest in the x-ray scattering measurement. In my role as reviewer, I have investigated the literature on roaming reactions and find the description in the manuscript inadequate. Specifically, the definition of roaming as a, "Reaction yielding products via reorientational motion in the long-range region of the potential," is incomplete. My understanding is that roaming involves unimolecular reactions where reactants and products are not separated by a narrowly-defined (specific molecular geometry) transition state, but rather a reaction where roughly flat regions of the free energy surface are critical to the reaction mechanism. For the study of Tarnovsky and co-workers, they claimed via simulation that wandering of the light component of the molecule (CH for CHBr₃ in Fig. 3 Nature Chem. 7, 562 (2015)) is central to the roaming reaction mechanism, in addition to the migration of a Br atom to form the new Br-Br bond of the isomer. This brings up two questions regarding the Bil3 investigated in this study.

→ It is certainly true that, as the reviewer pointed out, since its first report (*Science* 306, 1158, 2004), a roaming reaction is usually defined as a reaction pathway on a quite flat region of potential energy surfaces deviating from the minimum energy path (MEP) through a tight transition state. However, subsequent studies raised the need to define the roaming reaction with a more extended concept. Since the discovery of roaming in H₂CO, there have been many other examples of gas phase unimolecular dissociation as well as bimolecular reactions in which molecular products are formed directly via roaming.

Studies regarding roaming reactions have found the existence of saddle points on the quite flat roaming region, called "roaming saddle points", located separately from the region of the conventional saddle points (*J. Phys. Chem. Lett.* 2, 831, 2011; *Annu. Rev. Phys. Chem.* 62, 531, 2011). It has been found that the MEP from the roaming saddle points leads to the molecular products and their intrinsic reaction coordinates show similar features to roaming trajectories (*Molecular Physics* 112, 2516, 2014). For the dissociation reaction of NO₃, for example, it has been shown that the roaming pathway through roaming saddle points can act as a MEP since all of the molecular products are formed via roaming (*Science* 335, 1075, 2012, *J. Am. Chem. Soc.* 137, 3433, 2015).

Considering the new aspects of roaming reaction, Klippenstein et. al., defined a roaming reaction in their article (*J. Phys. Chem. A* 115, 14370, 2011) as follows: "We define a roaming reaction as one where orientational dynamics (roaming) in the long-range region of the potential (e.g., 2 – 4 Å separation where dispersion, electrostatic, hydrogen bonding, and/or ion-molecule forces are comparable to chemical forces) leads to a set of products that is different from those expected by simple continuation of the incipient

molecular decomposition (or alternatively of the incipient bond formation or abstraction if considering a bimolecular reaction instead of a unimolecular decomposition). In essence, we consider a roaming reaction to simply be one that produces an alternative set of products via reorientational motion in the long-range region of the potential. With this definition, the presence or absence of an alternative tight transition-state pathway to closed shell molecular products is irrelevant.”

In a recent review article by Suits (*Annu, Rev, Phys, Chem.* 71, 77, 2020), a more generalized definition of a roaming reaction, to which we referred in our work, is suggested as follows:

“A roaming reaction is one that yields products via reorientational motion in the long-range (3 – 8 Å) region of potential”

In addition to the roaming in the gas phase reactions, a recent work by Tarnovsky and co-workers (*Nat. Chem.* 7, 562, 2015) demonstrated that the roaming reaction can occur in the liquid solution phase as well, which is the first observation of roaming in solution. Since the reactions in the solution phase occur in a relatively more confined space than in the gas phase under the dielectric as well as mechanical influences of surrounding solvents, new features can possibly be observed for the roaming reaction in the solution phase and therefore we believe that the definition of roaming reaction still has room to be expanded even further in the future.

In summary, while we respect the reviewer's definition of the roaming reaction, which is a bit tight, we would like to point out that we adopted the most up-to-date definition, which covers a wider range of reactions. Regarding the reviewer's comment, “*For the study of Tarnovsky and co-workers, they claimed via simulation that wandering of the light component of the molecule (CH for CHBr₃ in Fig. 3 Nature Chem. 7, 562 (2015)) is central to the roaming reaction mechanism,*” please see our response to the next comment.

First, small and lower mass fragment have been central to the other roaming reactions investigated, but this is not consistent with BiI₃. Which molecular fragment would be roaming/wandering for BiI₃?

→ While all fragments and the atoms in them move with respect to each other, from the lab frame, the roaming reaction is often described in terms of the roaming movement of the smaller fragment or a lighter part of a fragment, which naturally has a higher speed. In other words, the part with the highest speed appears to move dominantly whereas the rest appears to have smaller movements. For example, in the study of Tarnovsky and coworkers (*Nat. Chem.* 7, 562, 2015) in which XBr₃ where X = CH, P, B or Al, they compared the roaming in XBr₃ to that of H₂CO. In H₂CO, the prominent orbiting motion of the H atom governs the roaming trajectory. Meanwhile, the motion of heavier Br is slower than H and restricted by the two remaining bromines and as a result the wandering of the central atoms is relatively more pronounced. The motion of Br is, however, not small, and both the Br and central atom are described to ‘roam’ around each other: “*As the central X atom wanders and recoils, Br(2) slowly roams further away from the Br(3)–X–Br(1) plane, with the partially dissociated X–Br(2) bond through a CI.*”

Thus, the roaming motion is not generally restricted to a particular fragment but rather can be described as a relative motion to each other, which can sometimes be predominated by a specific fragment depending on the molecular environment. In the case of BiI₃, the central atom (Bi) is much heavier to show prominent motions and therefore a partially dissociated iodine atom of relatively light weight than Bi appears to roam within the

restricted region by the two remaining iodines. This is why we stated the following in the previous manuscript.

“...in the case of BiI_3 , the wandering motion is mostly predominated by the iodine atom rather than the heavier BiI_2 fragment.”

Secondly, such a reaction mechanism should lead to significant increases in the distribution of bond lengths during the roaming reaction prior to bond isomer formation, but no evidence of increased distributions (whether it be a time dependent Debye-Waller or more sophisticated presentation of the distribution) is discussed. The issue of the distribution in bond lengths is addressed for the dissociation channel, but not the roaming reaction.

→ Considering the usually proposed flat potential energy surface for the roaming reaction, we also believe that it is plausible that during the roaming reaction, the relevant bond lengths or internuclear distances can have a wide distribution. With this said, we also note that the early isomer, which is the product of the roaming reaction, forms within 100 fs. Therefore, one possibility is that the current time resolution available at the XFEL is not sufficient to capture such motions of stochastic movements which might result in a wide distribution of internuclear distances, but resolves only the mean trajectory following a net direction of the movements.

Regarding this issue, we fitted the data again, with including Debye-Waller factors, $\exp(-0.5 \cdot q^2 \cdot \sigma^2)$ for the early isomer, where σ is the root-mean-squared displacement. Since the position of partially dissociated iodine, I(c), is expected to have a broad distribution, we applied a Debye-Waller factor to the atomic pairs containing I(c) exclusively. The fit results provide a σ value of ~ 0.1 Å for the atomic pairs containing I(c). This value is comparable to our error range for the structural parameters that can modulate those atomic pairs. The Bi-I(b)-I(c) angles, for example, have an error range of ± 3 degrees. This much of error in turn results in ± 0.14 Å errors for the distance of atomic pairs containing I(c), which covers comparably wider distance distributions to those generated based on the obtained σ . Thus, the distributions in pair distances during the roaming reaction of BiI_3 might be wide, but they are in the comparable range with the structural error range of our signal. Therefore, one possibility is that only the changes of the mean structure of BiI_3 along the net direction of the roaming reaction are observed through our experiment whereas other stochastic movements are not resolved within the structural error range as well as temporal resolution of our signal.

In this sense, the reviewer's concern actually renders an important subject of observing fine trajectories during roaming reaction, and we believe this will be resolved with a better experimental condition in the future.

Reflecting on the result on the σ value for the roaming reaction, we now added the following discussion to the revised manuscript.

“We note that introducing DWFs on the atomic pairs containing the roaming iodine atom (I_c) gives σ values of around 0.1 Å, which are comparable to the structural uncertainties of the distances for those atomic pairs. Therefore, one possibility is that only the changes of the mean structure of BiI_3 along the net direction of the roaming reaction are observed through our experiment whereas other stochastic movements involved in the roaming reaction are not resolved within the structural error range as well as temporal resolution of our signal. Regarding the obtained σ values, more details are given in the Supplementary Information.”

We also added a detailed procedure related to the fitting with Debye-Waller factors to the SI of the revised manuscript.

The second area that must be strengthened is the justification for the structural and kinetic models in the manuscript. Regarding the structural modeling, the isotropic difference scattering signal measures changes in the pair distribution function and is not specifically sensitive to bond angles or the three dimensional geometry of the solute. (As an aside, the anisotropic difference scattering is sensitive to bond angles, but this signal has not been analyzed in the manuscript. An explanation for why the authors chose not to analyze the anisotropic signal would be welcomed by this reviewer.) This means the x-ray scattering measurement in isolation underdetermines the three dimensional structure because there will be many ways of arranging the four atoms of BiI₃ that give the same pair distribution function, but different bond angles.

→ We respectfully disagree with the reviewer's statements about the relationship between the scattering signal and its sensitivity to the bond angles or the three dimensional geometry of the solute. The reviewer says that "*Regarding the structural modeling, the isotropic difference scattering signal measures changes in the pair distribution function and is not specifically sensitive to bond angles or the three dimensional geometry of the solute.*" It is correct that the isotropic scattering signal is indeed sensitive only to pair distribution functions. Nevertheless, because the scattering signal is sensitive to all possible pairs of atoms weighted by the number of pairs, it is sensitive also to bond angles as well. A triatomic molecule, for example a gold trimer complex, can be a good example (Nature, 582, 520, 2020). Since the scattering signal is sensitive to R(Au(1)-Au(3)) as well as R(Au(1)-Au(2)) and R(Au(2)-Au(3)), the bond angle Au(1)-Au(2)-Au(3) can be determined as well as the bond lengths.

The reviewer also states that "*...because there will be many ways of arranging the four atoms of BiI₃ that give the same pair distribution function, but different bond angles.*". Indeed, there will be many ways that give the same pair distribution function for a pair of Bi and an I atom, but the pair distribution functions for pairs of various combinations of I atoms would change. In other words, there are NOT many ways of arranging the four atoms of BiI₃ that give the same pair distribution functions for ALL pairs. In the following figure (Figure R1), as an example, we show various difference scattering curves generated by modifying one or all combinations of the three angles, I(a)-Bi-I(b), Bi-I(b)-I(c), and I(a)-Bi-I(b)-I(c), between the ground state and the early isomer of BiI₃.

Figure R1. (a-c) Ten difference scattering curves generated by modifying (a) I(a)-Bi-I(b) angles from 90 to 105 degrees, (b) Bi-I(b)-I(c) angles from 35 to 90 degrees, and (c) I(a)-Bi-I(b)-I(c) dihedral angles from 50 to 90 degrees. (d) 1000 difference scattering curves generated by all combinations of the three angles. As we can see in the figures, even though the signal dimension is reduced to a one-dimensional (isotropic) curve, the difference scattering curve sensitively changes its shape depending on the three-dimensional structure of the molecule. At least in the range of structural parameters that cover the possible geometries between and around the ground state and isomer of Bil₃, structures having different bond angles cannot give an identical difference scattering curve on a quite broad q -range of 1.0 ~ 6.5 Å⁻¹ or the same pair distribution function.

Regarding the anisotropic signal, the anisotropic signal from the Bil₃ solution is non-zero only around time zero and is identical to the anisotropic signal from a dye solution. This result means that the anisotropic signal is dominated by the Kerr effect of the solvent and for this reason, we chose not to analyze the anisotropic part of the difference scattering signal as there are no meaningful information to analyze for the dynamics of the solute (Bil₃) due to its negligible contribution to the anisotropy component. We explained it in the previous manuscript as follows and also showed the related figure in the Supplementary Information of the previous version of manuscript.

“Comparing the anisotropic contributions of difference scattering signals of Bil₃ solution to those of a dye solution which contain transient signals only from solvent response shows that they are identical, as shown in Supplementary Fig. 2. This result means a negligible contribution from Bil₃ solute molecules to the anisotropic component of the scattering signals.”

If the reviewer is curious about the origin of the vanishingly weak contribution of the Bil₃ molecule to the anisotropic part of the signal and actually asking if the time-dependent features of the anisotropic signal from the Bil₃ solution can be theoretically modelled, we believe that this is beyond the scope of the present work, and it will be reported in the future.

The manuscript needs to more clearly explain how the quantum chemistry structures are using in the structure refinement and how this constrains the search space for the data analysis.

→ We note that the purpose of the quantum calculation for the early isomer has been only for supporting the experimental finding, that is the structure of the early isomer, and characterizing its nature of electronic state. In other words, in the structure refinement for the early isomer, we relied on, without involving the quantum calculation results, free-parameter fitting process for the experimental signals by modifying the Bi-I bonds (I(a)-Bi and Bi-I(b)), Bi-I(b)-I(c) angle, I(a)-Bi-I(b)-I(c) dihedral angle, and I(a)-Bi-I(b) of BiI_3 through five structural parameters as described in SI section 5~7. Following is the related statement for this in our previous manuscript.

“To support our experimental observation for the early isomer of BiI_3 , we performed quantum chemical calculations using density functional theory (DFT), and a similar structure to that obtained from the experimental data was found in the triplet state (T_1). The DFT-optimized structure of the early isomer is shown in Supplementary Fig. 7 and is compared with the optimized structure from the global fit analysis on the experimental data.”

Thus, the quantum chemistry structure was not used in the structure refinement nor constrains the search space for the data analysis. To avoid any confusion, we added the following sentence to the relevant part.

“We note that the structure of the early isomer was obtained purely via the free-parameter fitting of the structural parameters, without the help of quantum calculation, during the GFA as detailed in the Supplementary information”

Regarding the kinetic modeling of the experimental findings, I have questions about the uniqueness and robustness of the analysis and require better evidence that the modeling and analysis is consistent with the claim a roaming reaction is occurring. First the SVD of the data and the structural modeling. The legitimacy of treating the time dependent signal as a sum of right singular vectors that can be associated with specific intermediates or products requires the geometric structure of those intermediates/products to be time independent.

→ Regarding the reviewer’s concern of the treatment of time-dependent signal with the time-independent structure of intermediates or products through SVD, we note that those time-independent structures are extracted from a reconstructed signal based only on the first two singular vectors rather than from the whole experimental signal. Pages S4 ~ S10 in Supplementary Texts of the previous manuscript are relevant to this question.

As evidenced through the Supplementary Fig. 14 (or Supplementary Fig. 16 for the revised Supplementary Information), the shifts in the scattering signal during the first picosecond, which we have interpreted as a signal of time-dependent structural dynamics, mostly come from the 3rd and 4th singular vectors and those signals, as pointed out by the reviewer, should not be treated with the time-independent geometry of intermediates or products. Analysis based on the two separate parts of signals can be rationalized through the following equations, which are now added to the revised Supplementary Information.

$$\Delta S(q, t) = c(t) * \{S^{EZ}(q, t) - S^{EZ}(q)\} + \Delta S_{\text{solvent}}(q, t) \quad (\text{S3})$$

$$= c(t) * [\{S^{EZ}(q, t) - S^{EZ, \text{EQ}}(q)\} + \{S^{EZ, \text{EQ}}(q) - S^{EZ}(q)\}] + \Delta S_{\text{solvent}}(q, t) \quad (\text{S4})$$

$$= [c(t) * \{S^{EZ}(q, t) - S^{EZ, \text{EQ}}(q)\}] + [c(t) * \{S^{EZ, \text{EQ}}(q) - S^{EZ}(q)\} + \Delta S_{\text{solvent}}(q, t)] \quad (\text{S5})$$

$$= \Delta S_{\text{residue}}(q, t) + \Delta S'(q, t) \quad (\text{S6})$$

Briefly, the difference scattering signal can be considered as the sum of (i) the fractional concentration multiplied by the difference scattering arising from the change of the excited state (es) structure with respect to the equilibrium (eq) structure of the es and (ii) the fractional concentration multiplied by the difference of scattering signal between the eq structure of the es and that of the ground state (gs) plus the difference scattering signal arising from the bulk solvent due to its temperature change ($\Delta T(t)$, omitted in the equations for simplicity), as shown in equation (S3). The signal component (ii), which is $\Delta S(q, t)$ in the equation (S4), corresponds to the reconstructed signal from the first two singular vectors, and the time-dependency of $\Delta S(q, t)$ comes only from the fractional concentration, $c(t)$, and the temperature of solvent, $\Delta T(t)$. Thus, the fractional concentration profile of the es and its time-independent equilibrium structure as well as the time profile of the temperature of the bulk solvent can be obtained by analyzing $\Delta S(q, t)$. The corresponding procedures are described in pages S7 ~ S11 in Supplementary Texts of the revised Supplementary Information. As a next step, the time-dependent structural motion can be obtained by analyzing $\Delta S_{\text{residue}}(q, t)$ or, equivalently, the whole signal, $\Delta S(q, t)$, based on the information of kinetics and equilibrium structure obtained from $\Delta S(q, t)$. We chose the later method, and the corresponding procedures are given in pages S11 ~ S15 in Supplementary Texts of the revised Supplementary Information.

The clear shifts in the scattering signal during the first picosecond would indicate that any kinetic model would be dubious for the first picosecond.

→ Following the formalism in equations (S3)~(S6), the kinetic model was determined in the analysis of $\Delta S(q, t)$ which does not incorporate the shifting feature during the first picosecond shown in the whole scattering signal. Subsequent analysis for the whole experimental data, the ultrafast signal feature is then reproduced through the structural motion of reacting species under the predetermined kinetic model. Thus, a wrong kinetic model will not give a satisfactory fit on the reconstructed data, $\Delta S(q, t)$, and in turn leads to failure in structure fitting for reproducing the ultrafast signal feature in the whole experimental signal, $\Delta S(q, t)$.

I also question the need for four exponential time constants in the measurement. Including the error bars, the 8.8 ps and 11.9 ps time constants nearly overlap, particularly considering the fit shown in Fig. S13 appears to have an offset.

→ If only three exponentials are used, the fit result is not satisfactory at all as shown in Supplementary Fig. 13(c) of the previous version of manuscript (Supplementary Fig. 6(b) for the revised Supplementary Information). This fact was stated in the Supplementary Information, and is also mentioned in the main text of the revised manuscript. Moreover, if only three exponentials are used, then the whole kinetic framework of the solute needs to be described with only two exponentials (one is for the solvent heating). Such a kinetic framework cannot explain the experimental data well. To further unroot any concern regarding this point, we added the following analysis. In the following, two possible kinetic frameworks C and D which use the two exponentials for the solute kinetics are shown.

Figure R2. (a) Possible kinetic frameworks using the two exponentials for the solute kinetics. (b) Relative reduced chi-square (χ_{red}^2) values obtained from global fit analysis based on each of kinetic models built from the two frameworks in (a) with respect to the minimum χ_{ed}^2 value obtained from that based on the kinetic model A-3. (c) Resultant fit obtained from the kinetic model D-1 in comparison with that obtained from kinetic model A-3 at late time delays.

When three exponentials are used, time constants of 577 fs, 3.68 ps, and 7.52 ps are obtained. Among them we assign the 577 fs to the solvent heating as it is in the similar timescale of the best kinetic model proposed in our manuscript. Then the remaining two time constants can be used to build the kinetics of solutes based on the kinetic model frameworks C and D. Distributing the two time constants to the model C and D leads to 4 possible kinetic models denoted here as C-1, C-2, D-1, and D-2. Following the same analysis procedures described in our manuscript, we extracted $\Delta S(q, t)$ and analyzed it by fitting the equilibrium structure of X (the early isomer) and branching ratios for each of the 4 kinetic models. The resultant relative reduced chi-square (χ_{red}^2) values with respect to the minimum χ_{ed}^2 value obtained from kinetic model A-3 (the kinetic model of Fig. 3(a) in the manuscript) is summarized in the Fig. R2(b).

The fit quality becomes worse compared to the fit based on the kinetic model of Fig. 3(a) as can be seen through the relative χ_{red}^2 larger than one meaning a larger χ_{ed}^2 than that obtained from the suggested model shown in Fig. 3(a). In particular, the simplified kinetic model does not provide satisfactory fit at late time delays as can be seen in Fig R2(c).

We note that there are 12 possible kinetic models if 577 fs is allowed to be responsible for the solute kinetics, but the conclusion is the same. We added this discussion to the Supplementary Information of the revised manuscript.

“We note that if only three exponentials are used, then the whole kinetic framework of the solute needs to be described with only two exponentials (one is for the solvent heating). Such a kinetic framework cannot explain the experimental data well (the details and fits are not shown).”

Lastly and most importantly, a model for a roaming reaction that involves the time dependent rise in an intermediate isomer with a fixed structure does not possess the essential dynamics

of a roaming reaction. Where is the roaming/wandering? I fully appreciate the significant challenges associated with such modeling, but without it where is the evidence in support of the primary conclusions from the manuscript?

→ Again, one evidence for the roaming-mediated reaction is that a molecular product, the early isomer, is formed within 100 fs through the dynamic movement of an incipient iodine atom that is partially dissociated from Bi. This time scale is much faster than that for the collision of the dissociated I atom and the solvent cage. Such an ultrafast formation of the early isomer was also used as the supporting evidence for the roaming-mediated reaction in the work of Tarnovsky and coworkers (*Nat. Chem.* 7, 562, 2015). The reviewer's concern boils down to whether there is any structural evidence in terms of the actual time-resolved motion of atoms in the formation of the early isomer. This information can be found in the time-dependent Bi-I(b)-I(c) angle and I(a)-Bi-I(b)-I(c) dihedral angle, especially in the time delays of 0 ~ 100 fs, as shown in Fig. 4(a).

Nevertheless, the reviewer's concern regarding this point prompted us to show plots comparing the Bi-I(c) distance, the I(b)-I(c) distance, and the Bi-I(c)-I(b) angle for the dissociation pathway and the isomer-formation pathway. These plots are shown in a new figure (Figure 5). For this, we improved the fitting process and we note that the plots in Figures 2 ~ 4 have also been updated based on the new fitting results although the changes would not be noticeable in the current scale.

Fig. 5. Comparison of the structural motions for the dissociation and roaming-mediated isomerization channels. **a.** Bi...I_c distance, **b.** Bi-I_b-I_c angle, and **c.** I_b...I_c distance for the BiI₃ in the dissociation pathway (black) and the roaming-mediated isomerization pathway (red) are shown, and the error bars represent the one-standard-deviation error values obtained from the relevant structural parameters from the fit.

These comparative plots highlight the roaming nature reflected on these structural parameters and the clear characteristic of the frustrated fission of the Bi-I(c) bond for the roaming-mediated isomer formation. A paragraph regarding this point was also added to the revised text as follows, and the description about the structural fitting process was updated in Supplementary Information.

“Finally, comparisons of the relevant structural parameters, the Bi - - - I_c distance, the I_b-I_c distance, and the Bi-I_b-I_c angle, for the dissociation pathway and the roaming-mediated isomer-formation pathway vividly visualize the different motions of the atoms depending on

the pathway (Fig. 5). For both pathways, the Bi - - - I_c distance initially follows a similar path up to ~100 fs (Fig. 5a). Then, the Bi - - - I_c distance stops increasing for the isomer-formation pathway, demonstrating the nature of the frustrated fission of the Bi - - - I_c bond for the roaming-mediated reaction whereas it keeps increasing for the dissociation pathway. In contrast, the Bi-I_b-I_c angle shows a substantially faster increase up to ~100 fs and a saturation afterward for the isomer formation whereas its increase is relatively marginal throughout the whole time scale for the dissociation pathway (Fig. 5b). The contrast between dissociation and the roaming-mediated isomer formation is even clearer in the I_b - - - I_c distance, which keeps increasing for dissociation whereas it initially decreases up to ~100 fs for isomer formation (Fig. 5c).”

Responses to the comments from Reviewer #2

This is a study on the dissociation and isomerization dynamics and kinetics of bismuth triiodide (BiI₃) in acetonitrile solution. The authors used time-resolved X-ray scattering with femtosecond temporal resolution to measure the intramolecular structural rearrangements of the molecule upon optical excitation. They find a roaming-mediated isomerization process via a hitherto undetected transient intermediate (“early isomer” in a triplet state) and they relate this interesting process to parallel pathways (dissociation and geminate recombination) and to pathways that the same authors or part of the authors revealed in an earlier study at a synchrotron radiation source on slower time scales.

I find this a very convincing study and I think that the data and presentation are just wonderful. The manuscript is very clear, the results are exceptionally clear-cut and the discussion is very focused and of outstanding accessibility.

While solvent cage-induced isomerization in similar molecules is thought to proceed on time scales of several hundreds of femtoseconds and beyond, where collisions with solvent molecules induce radical recombination, here the authors clearly detect a new mechanism. Within about 100 fs they find the direct formation of an “early isomer”, a time scale that is clearly much shorter than typical time scales of solute-solvent interactions. They very convincingly show how this isomer results from (coherent) nuclear motions directly induced by the electronic excitation and how it then proceeds to the final isomer (or back to the initial ground state in a parallel pathway). The authors identify a particular I atom in the system that does not quite dissociate (“frustrated bond fission”) and instead “wanders around”, trapped in the solvent cage in the vicinity of the remaining fragment. At the same time, this “wandering around” happens so fast (in about 100 fs) that it is too fast for the solvent cage to participate or even determine the motions of the solute atoms. A fascinating case, where comparably long-range interactions within an “almost dissociated molecule” in solution (at distances of 3-4 Å) lead to very interesting molecular transformations.

I find the level of detail to which the authors followed nuclear motions in time and to which they characterized the structure of the essential reaction intermediate (Figures 3+4) very fascinating. The system, consisting of comparably heavy atoms for good X-ray scattering signals, is well chosen and wonderfully serves the purpose of distinguishing numerous parallel reaction pathways and identifying the one that is really interesting, which the authors rightfully term “roaming-mediated isomerization”. This is an excellent demonstration of the capabilities of the employed technique and it is a very convincing demonstration of the importance of “roaming reactions” in solution, a topic that has increasingly gained attention in recent years. It will be very interesting to see how such measurements can be used in the future to see even finer details such as, maybe, how solute-solvent interactions do interfere with the observed intramolecular roaming process by manipulating the potential energy landscape. Certainly, a great step towards understanding photochemical transformation on the atomic length and time scales.

→ The authors appreciate the positive evaluation of our work. In the following, we tried to address the reviewer’s remaining concerns.

Introduction, p. 3 “indicating that roaming reaction is working in the liquid solution phase as well”. Given the text-book character of the work which should be very accessible to graduate students as well, I wonder whether the authors could add 1-2 sentences on why (they think that) roaming reactions in solution are thought to be unusual. Is that because they are hard to observe or are there conceptual arguments? Later in the same paragraph they mention a “flat region of the S1” potential energy surface and maybe this is a conceptual framework in which what I consider missing could be formulated.

→ For chemical reactions involving dissociations in the liquid solution phase, the resultant radicals or ions can interact with the surrounding solvents, influencing the subsequent formation of molecular products. Thus, the kinetics and yields of the molecular products are strongly affected by the properties of solvents. The roaming pathway in solution provides a very efficient and direct channel to form the molecular products without being perturbed by the solvents. In this regard, the roaming reaction is unusual because it occurs as in a gas-phase reaction in the presence of a solvent. Following the reviewer’s suggestions, we added the following sentences in the revised manuscript to stress the uniqueness of the roaming in the solution phase.

“Thus, these findings indicate that through the roaming on the flat region of the potential, an efficient formation of molecular products such as an isomer can occur upon a photofragmentation reaction in the solution phase as in the gas-phase.”

P. 5, “As a result, we obtained four time constants...” Understanding that this may be hard or impossible (?), can the four time scales, or some of them, even if indirect or qualitative, be related to (some of) the experimental observables in Figure 2? A few sentences here could help to better relate the structural and kinetic information with what was actually measured.

→ We modified the part referred to by the reviewer as follows.

“As shown in Supplementary Fig. 6a and b, a sum of four exponential functions fit those RSVs satisfactorily, whereas a sum of three exponential functions did not. The four time constants were determined to be 508 (± 13) fs, 3.11 (± 0.43) ps, 8.83 (± 1.11) ps, and 11.90 (± 1.67) ps, and the full width at half-maximum (FWHM) of the IRF was determined to be 162 (± 7) fs.”

We also modified the part relevant with the assignment of the four time constants to the experimental observables in Fig. 2(d) as follows.

“In the best kinetic model, the first time constant, 508 fs, is assigned to the kinetics of the solvent, and the remaining three constants to the solute kinetics. The best kinetic model and the corresponding time-dependent concentrations of reacting species are shown in Fig. 3a and b, respectively and the kinetics of bulk solvent temperature, whose time constant is 508 (± 13) fs, is shown in Supplementary Fig. 9.”

Figure 1: I understand that this is a figure that is supposed to motivate the present study instead of summarizing the inferred reactions pathways. Still, maybe the figure and its content can be enhanced by adding time scales that are known from other studies (such as their previous study referenced in ref. 18)? Also, I have a hard time “seeing” the first process/scenario in the blue inset (“isomerization via a bimolecular recombination”). Maybe it is possible to include both, this and the other scenario of a roaming mediated reaction in the figure to further enhance how meaningful and far-reaching the detection of the roaming

mechanism is? Would, e.g., bimolecular recombination happen via a new pathway that would connect the depicted pathways along $R_{Bi...I}$ and along θ_{Bi-I-I} ?

→ We appreciate the constructive suggestions for Figure 1. The new Figure 1 modified according to the comments is shown below.

Figure 3: Can, in panel (a), the time scales for the very first steps (50% and 50%) be indicated (even if these may happen within the temporal resolution of the experiment)? From the four time constants mentioned in the text (see above), the one with 500 fs is missing in this figure and it may be instructive to add it to the figure caption with a short note that it relates to solvent heating instead of solute intramolecular structural rearrangements. In panel (c), can the I atoms be numbered 1-3 in the top and bottom panels to better “see” the different views of a given structure (up/down) and to better follow the very nice transformations of the molecules (left/right)?

→ We appreciate the reviewer’s comments, which we believe help enhance Figure 3 greatly. We modified the figure caption to mention the time scale for solvent heating as follows.

“The time constant of 508 fs which is missing here is assigned to the kinetics of solvent.”

New Figure 3 modified according to the comments is shown below.

Responses to the comments from Reviewer #3

The paper is an interesting and convincing study of a simple isomerization reaction in solution. So it is an important step forward in this active field and I recommend publication.

I do think a few relevant papers need to be cited.

Photodissociation dynamics of nitromethane and methyl nitrite by infrared multiphoton dissociation imaging with quasiclassical trajectory calculations: Signatures of the roaming pathway, Dey et al. J. Chem. Phys. 140, 054305 (2014);

Ultrafast Proton Transfer Dynamics on the Repulsive Potential of the Ethanol Dication: Roaming-Mediated Isomerization versus Coulomb Explosion, Wang et al J. Phys. Chem. A 2020, 124, 14, 2785

Capturing roaming molecular fragments in real time, Endo et al, Science 370, 1072 (2020)

Also some real time roaming papers from the Dantus group.

→ The authors appreciate the positive evaluation of our work. In the revised version, we added citations to the following papers, which include the suggested references.

Photodissociation dynamics of nitromethane and methyl nitrite by infrared multiphoton dissociation imaging with quasiclassical trajectory calculations: Signatures of the roaming pathway, Dey et al. J. Chem. Phys. 140, 054305, 2014.

Ultrafast Proton Transfer Dynamics on the Repulsive Potential of the Ethanol Dication: Roaming-Mediated Isomerization versus Coulomb Explosion, Wang et al J. Phys. Chem. A, 124, 2785, 2020.

Capturing roaming molecular fragments in real time, Endo et al, Science 370, 1072, 2020.

Substituent effects on H_3^+ formation via H_2 roaming mechanisms from organic molecules under strong-field photodissociation, Ekanayake et al, J. Chem. Phys. 149, 244310, 2018.

H_2 roaming chemistry and the formation of H_3^+ from organic molecules in strong laser fields, Ekanayake et al, Nat. Commun. 9, 5186, 2018.

Roaming, Bowman, Molecular Physics 112, 2516, 2014

Roaming dynamics in radical addition/elimination reactions, Joalland et al., Nat. Commun. 5, 4064, 2014

Figure 1 is indeed highly schematic and unfortunately misleading. Real wavepackets on anharmonic, reactive potentials do not maintain simple Gaussian shapes; quite the opposite, e.g., major breakup into packet pieces. So I strongly suggest dropping those imagined wavepackets and just use arrows.

→ We greatly appreciate the reviewer's careful attention and suggestions. New Figure 1 modified according to the comments is shown below.

✓ Roaming-mediated direct isomerization or cage-induced isomerization?

✓ Wavepacket trajectory for the isomerization

✓ Wavepacket trajectory of the dissociation fragments

REVIEWERS' COMMENTS

Editorial Note: Reviewer 1 provided comments to the editor only, stating that they now support publication of the work.

Reviewer #2 (Remarks to the Author):

The authors submitted a revised version of their manuscript with changes to the main text and figures and additional supporting information. My concerns, which were marginal anyway, have been well addressed in the revisions. Roaming reactions, still being loosely defined or obscure and new to some extent, are now addressed more specifically in the text and, from my perspective, as specifically as the topic may allow. Also, relation of the present findings to previous work (including longer time scales than probed here, see Figure 3) are better visible. I can, still and again, recommend publication of this manuscript as is.

Responses to the comments from Reviewer #2

The authors submitted a revised version of their manuscript with changes to the main text and figures and additional supporting information. My concerns, which were marginal anyway, have been well addressed in the revisions. Roaming reactions, still being loosely defined or obscure and new to some extent, are now addressed more specifically in the text and, from my perspective, as specifically as the topic may allow. Also, relation of the present findings to previous work (including longer time scales than probed here, see Figure 3) are better visible. I can, still and again, recommend publication of this manuscript as is.

→ We thank the reviewer for supporting the publication.